# Success Factors Influencing Peer-to-Peer Lending to Support Financial Innovation

**Natnara Chulawate**  **and Supaporn Kiattisin \***

Faculty of Engineering, Mahidol University, Nakhon Pathom 73170, Thailand
* Correspondence: supaporn.kit@mahidol.ac.th; Tel.: +66-81-866-4207

**Abstract:** The purpose of this study is to identify success factors that are conducive to developing the ability to create financial innovation within developing countries for the sake of sustainability. The purpose of this research is to contribute to the identification of success factors. The case study involves a peer-to-peer lending (P2P Lending) business operator in Thailand and focuses on the lender perspective. The results consist of 13 potential factors driving financial innovation in process improvement. The study collected data from 300 respondents through a structured questionnaire. The structural equation model was used to analyze the data via Mplus version 7. In order to gain a better understanding, we emphasize that each country's financial business may show different success factors due to different situations and environments, which might pose a challenge when drawing conclusions from the survey and building sustainability in the financial industry. The research summarizes the factors of success in 3 contexts with 13 factors; namely, the risk context consists of a higher interest rate, inflation increase, macroeconomics, regulation laws, and legal, liquidity, and finance and credit status. The trust context includes demographic characteristics, biological characteristics, and an individual's reputational capital, and the lender perspective information context includes loan delinquencies, funded loans, politics, and culture. According to our results, the investor or lender will benefit from bringing concepts and methods that involve adopting international loans.

**Keywords:** P2P lending; finance innovation; lender-centric; finance literacy; success factor; international loan

## 1. Introduction

Peer-to-peer lending is borrowing between individual borrowers and lenders and investors through online channels. The first P2P lending was established in 2005 by the British company Zopa. Subsequently, the platform gained popularity and grew in many countries, such as the United States, the European Union, China, etc. [1,2]. P2P lending is a financial innovation widely used in many countries. However, in Thailand, P2P lending platforms are still considered new and are not well-known. There are only 4–5 such platform providers in Thailand, which operate in a limited area. Regarding borrowers, primarily seek small business loans and personal car title loans [3].

The P2P lending platform function matches the borrower and the lender and arranges the credit contract with or without collateral. As a financial intermediary is used instead of a bank or a financial institution, the advantage of such transactions is that they can be requested by one borrower or loan applicant [2,4]. However, there can be more than one co-lender. At the same time, one lender can distribute several loans. As a result, P2P lending interest rates are higher than those of financial institutions. However, they are lower than the interest rates on informal loans, which can have very high-interest rates. The borrower or loan applicant has the opportunity to receive a rate offer. Interest rates are lower than when borrowing money from financial institutions. Additionally, lenders achieve higher returns than they receive on deposits or if they were buying government

bonds. P2P Lending is the use of technology to act as a financial intermediary. It increases the opportunity and options to access financial services for people and businesses who do not have access to borrowing or credit from financial institutions [5].

Funds are essential to human life as situations and problems arise. The larger the population, the greater the demand for consumption. Technology means that people have more finance literature. In the digital age, there are many sources of funding. P2P lending is a platform to obtain new capital [6]. By eliminating the problem of high-yielding mediators and unscrupulous lending, which endangers borrowers, P2P lending is a convenient and safe form of lending. Borrowing problems before P2P lending involve the borrowing gap between individuals [7]. P2P lending solves the high-interest rate risk. Evil and intense desires cannot be monitored and pursued. It can solve such problems to achieve fairness in borrowing; as a result, it reduces the borrower's risk of interest payments and the risks associated with debt collection.

Furthermore, P2P lending can build trust and give lenders and borrowers confidence in their loan transactions. A traditional loan is facilitated by a financial institution. The problem is that financial institutions only lend money. Those with a history in the credit bureaus only cause inconsistencies in borrowing. Therefore, P2P lending is the solution. Everyone has the right to borrow money through an online platform, As shown in Figure 1.

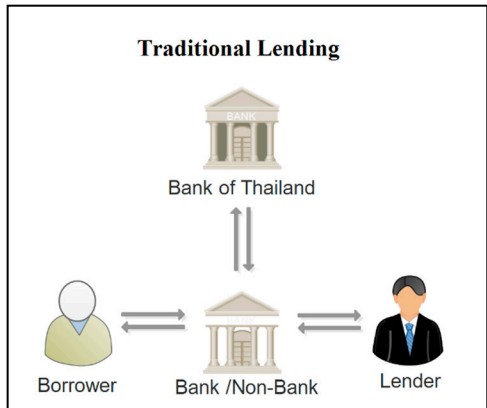
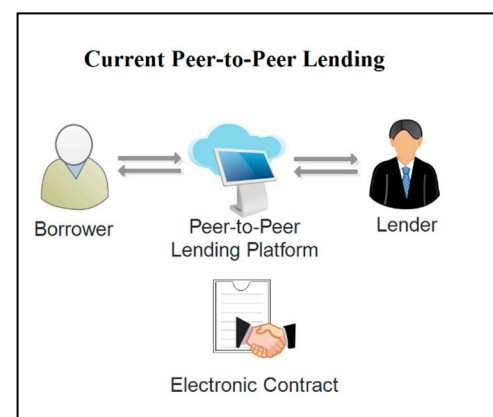

**Figure 1.** Traditional lending and current P2P lending.

This study brings a prototype of the P2P lending platform to help readers understand the system's structure and increase the visibility of this business. First, we recognize the importance and interest in the P2P lending system. Next, we generate a conceptual model for the success factors that have been identified. Failure factors involve elements from the lender's point of view [8]. The benefits of P2P lending platforms require further study so that these concepts and methods can be applied in the real world [9].

In summary, this paper shows that platform reliability is essential in platform selection, and other factors are involved, such as interest rates, inflation rates, and regulators. The study creates a conceptual model to find the elements used to help the decision-making of researchers [10,11]. Based on the principles of financial innovation, financial risk management services are an essential part of the success factor in decision-making for sustainable P2P lending services. The benefits for new investors and borrowers lie in generating knowledge and understanding the reliability of P2P lending systems. Furthermore, sustainability the online loan businesses and success factors can also be applied to international loans [12,13].

## 2. Literature Review

### 2.1. Success Factor Influencing Support for P2P Lending

This paper aims to create a conceptual model that is appropriate and able to close the risk gap and build credibility, including selecting P2P lending methods suitable to different

situations [14,15]. Therefore, we prepared a success factor influencing conceptual model by extracting factors and variables from previous research [16]. To close the traditional loan gap and improve the financial innovation process, we identified three contexts: risk factors, trust factors, and lender perspective information factors [17]. Furthermore, P2P Lending reduces the risk of default, which is the main problem for lenders and results in the loss of economic opportunities in other channels; therefore, the key topics we reviewed became this research's success factors. Thus, our findings group the contexts and describe them as shown in Table 1.

**Table 1.** Key topic success factors.

| Dimension/Context | Sub-Dimension | Number of Sub-Dimension | Description/Reference |
|---|---|---|---|
| Risk | higher interest rate | (1) | The interest rate is higher than the standard rate in the P2P lending process [14,15,18–24]. |
| | inflation increase | (2) | Inflation increases P2P lending [16,25]. |
| | macroeconomics | (3) | The study of the behavior of large economic units or the whole country, including national or global economic problems such as national income, GDP, GNP, finance, banking, international trade, economic development, savings and investments, labor, unemployment, and money supply [16]. |
| | regulation laws and legal | (4) | The system of rules that a particular country or community recognizes as regulating the actions of its members and that is enforced by the imposition of penalties [9,26–30]. |
| | liquidity | (5) | The ease with which an asset can be converted into ready cash without affecting its market price [31–34]. |
| | financial and credit status | (6) | Monetary receipts and expenditures [35]. |
| Trust | demographic characteristics | (7) | Characteristics of age, gender, education, and socioeconomic status of P2P lenders [10]. |
| | Biological characteristics | (8) | Activities that study living things, such as borrower activity [36]. |
| | individual's reputational capital | (9) | The individual value of all intangible corporate assets [37]. |
| Lender Perspective Information | loan delinquencies | (10) | The borrower breaches the contract and does not pay the debt as scheduled [8,38]. |
| | funded loan | (11) | The lump sum that financial institutions lend [39,40]. |
| | politics | (12) | Process and method will lead to the decision of the national group [41,42]. |
| | culture | (13) | The social behavior, institutions, and norms found in human societies [1,43,44]. |

The critical topic success factors occur in three contexts. The risk context contains the "higher interest rate", "inflation increase", "macro-economic", "regulation laws and legal", "liquidity", and "financial and credit status" factors. The trust context contains the "demographic characteristics", "biological characteristics", and "individual's reputational capital" factors. The lender perspective information context contains the "loan delinquencies", "funded loan", "politics", and "culture" factors. The details of the thirteen factors are listed in Table 1.

*2.2. Perceived Risks and Trust in P2P Lending*

Perceived risk measures P2P lending reliability via a physical examination. It is a communication tool for lenders to recognize the potential for fraud. For one thing, P2P lending creates a robust foundation, and it can cause investors to lose capital and cause investment anxiety [28]. Building confidence between borrowers and lenders when using P2P lending lead to a positive relationship and gives borrowers and lenders confidence in the finance system through a solid and attractive social network [29,30]. As a result, platforms are trusted to meet the platform's funding system verification needs [30].

Another factor that causes credibility among borrowers [31] is that the borrower's online information includes a photograph [32], age, gender, and identity card [33]. Unreliability in P2P lending only occurs when payment cycles are set far apart and over large geographic distances [34]. In conclusion, studies have shown that trust in social networks is required first. However, the risk is reduced after social media is used to assess the borrower's physical characteristics [35].

Regarding the failure factors of P2P lending, the platform company provides a platform to guarantee the investment, loan amount, and interest rate, and a low-interest rate will attract borrowers who are interested in borrowing. If the credit line has a high rate, it will also attract interested people. These can affect the lender's trust, causing the failure of P2P lending [36–45].

*2.3. Financial Innovation Management*

Financial innovation management needs to be improved. An innovative process involves a thorough analysis of financial activities, as well as economic and social development for financial sustainability. Financial innovation in each country has led to the development of innovative products and services that play an essential role in enhancing the quality of life of the people of that country. Ensuring that people are comfortable with transactions while considering the outcome of innovation is the most important aspect. Capital is the primary outcome of financial management [17]. This study provides a conceptual model that supports decision-making in P2P lending. We focus on identifying the success factors that lead to the success of financial innovation and investment considerations [34]. Although this study obtains funds on the part of lenders or investors, most research still breaks down these factors, which necessitates a focus on supporting innovation in online lending. Most analyses of different factors involved in risk and credibility only consider on one side [46].

This literature review involves the study of theories and concepts via identifying the needs of general users so that platforms can meet the needs of users as much as possible in order that developers can choose to cater to more general user needs. P2P lending necessitates the study of user behavior, user experience, and the expectations of platform users [47]. What makes online borrowing attractive is good P2P lending that must be able to control the credit risk of the loan to the lender. P2P lending reduces financial costs and eliminates the traditional financial system. The pricing strategy consists of a subscription fee and a per-transaction fee that can attract investors interested in being a P2P lender [48–50]. We aim to identify process improvements for financial innovation that can create a sustainable process for a particular country.

## 3. Methodology

Our methodology starts with a literature review to assess the importance of and gaps in P2P lending identified in previous research. Then, the factors and variables aware analyzed, and the factors are mapped. Then, we prepare a conceptual model to determine the research scope and craft a P2P lending application before distributing questionnaires and collecting data for analysis an evaluation [39]. Finally, we collect the results and discuss them. This process is shown in Figure 2.

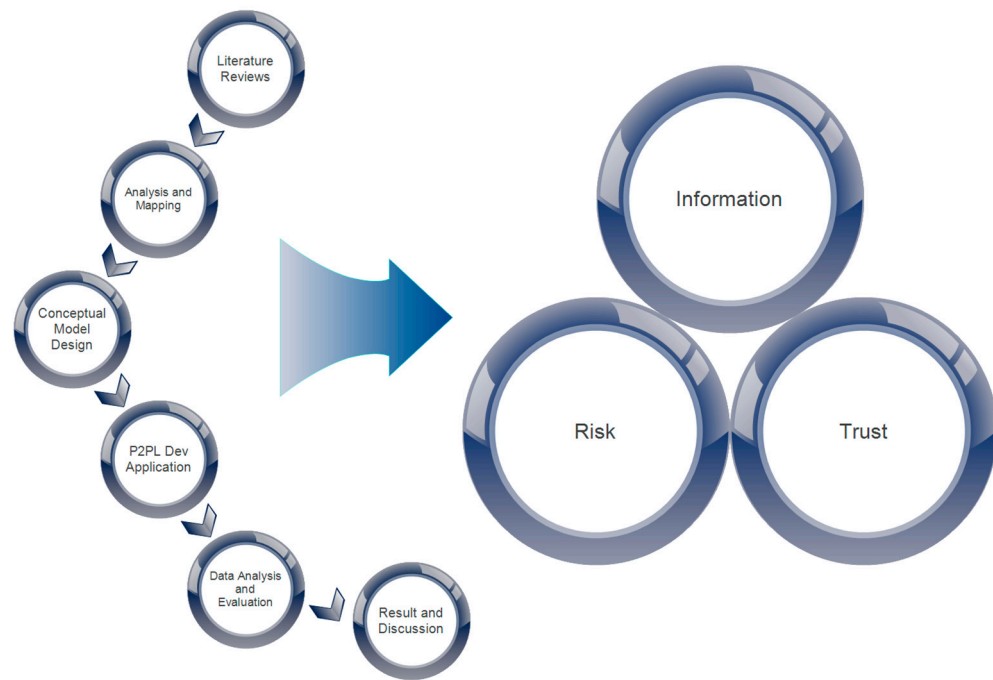

**Figure 2.** Research methodology.

### 3.1. Analysis and Mapping Process

The analysis and mapping process is based on extracting relevant literature review factors and dividing them into three groups: the risk context (higher interest rates, inflation increases, macro-economics, regulation laws and legal, liquidity risks, and the financial and credit status/income), the trust context (demographic characteristics, biological factors, and individual reputational capital), and the lender perspective information context (loan delinquencies, funded loan, politics, and culture). These groups are shown in Table 1. The extracted factors were used to create a conceptual model to determine correlations for a reliability analysis [51]. In the lender's view, risk tolerance includes the resulting evaluation model to build credibility in future P2P lending platforms [52].

### 3.2. Conceptual Model Design

The study divided P2P lending into the lender and borrower sides, where P2P lending connects lenders and borrowers. The research presents the lender's perspective because lenders incur risk when lending if the used P2P lending platform is unreliable or lacks expertise when it comes to credit scoring, which will cause lenders to have problems with loan delinquencies and lost financial opportunities. Therefore, the main objective of this research is to create a conceptual model that identifies the trust, risk, and lender perspective information factors of a P2P lending platform to close the consideration gap and increase lenders' confidence.

### 3.3. Peer-to-Peer Lending Platform Development Application

Our application development team has developed a P2P lending platform development plan in the form of a software development plan that is divided into the following sections: scope, analysis/software requirements, design, development, testing, unit testing, and link UX/UI. The system architecture and the project of developing a platform and a P2P lending system through a P2P lending web application has three main components [48,53–55].

The system's front-end service runs on a web application platform. Both the lender and the borrower use web application technology [56]. Connecting to parts of the service platform or an intermediary of the P2P lending system is the system's primary function

in the management of the processing, analyzing, linking, and verifying of the data. The process of knowing your customer (KYC), the credit rating of the borrower (creditworthiness), checking the facts surrounding the service user customer due diligence (CDD), and verifying the identity of the user are all involved in the lender suitability assessment system. Internal processing with artificial intelligence (AI) enables machine learning (ML), which can help automate the process.

Moreover, the application is linked to various systems for data verification and confirmation by connecting the application interface (application program interface: API) with relevant agencies such as the Department of Provincial Administration Bank of Thailand [57]. It also connects to a block chain network, a platform type. Decentralized applications (Dapps) are utilized in the process of creating financial contract documents. Alternatively, a loan agreement based on the system's reliability is one of the three basic principles of developing an intermediary platform or service intermediary that runs on the cloud (cloud server). Data management and control are carried out by the central platforms or the main admin section. The caretaker uses the data management system administration or a powerful platform, including various activities incurred in the interpersonal loan application process, for surveillance. We ensure the safety and build the confidence of service users, both the borrower and the lender [27,58], via the process and workflow of the P2P lending system involving a smart web application. The workflow and operation of the system consist of borrowers, lenders, and intermediary systems or platforms, which interact with details of the borrowing process. Lending and arranging developers can create a contract (borrowing and lending documents or loan agreements).

*3.4. Data Analysis and Evaluation*

3.4.1. Data Analysis

Phase 1: We identify the key success factors for managing an online loan platform in Thailand. The content validity index (CVI) measures the validity of the research tool. To measure the reliability of the questionnaire. The study used Cronbach's alpha to measure the questionnaire's reliability. This statistic reflects the relative relationship between all topics. The statistical value is between 0 and 1, and a score above 0.7 is considered reliable. The questionnaire was tested for validity and reliability before being sent to the lenders [48].

Phase 2: The study analyzes the critical success factors of online loan platform management in Thailand by designing a theoretical reference query (using confirmatory factor analysis CFA) to statistically confirm elements or indicators and decide whether they support the hypothesis [54,59,60]. The trust of lenders as measured by the CFA analysis method has five steps: parameter estimation, verifying model consistency, model fitting, and interpretation of the analysis results. The criteria for determining the conformity of the confirmation components consisted of models assessed using the Chi-squared goodness of fit ($p > 0.05$), the comparative fit index (CFI > 0.90), the goodness of fit index (GFI > 0.90), the root mean square error of estimates (RMSEA < 0.08), and the average variance extracted (AVE > 0.50) [61]. The AVE is an indicator of convergence; it refers to the mean variance extracted for the items loaded on a construct. Therefore, the average variance extracted (AVE) should be above 0.5. The AVE is derived from Equation (1) as follows:

$$\text{AV} = \frac{\sum \lambda^2}{n} \tag{1}$$

3.4.2. Population and Sample Size

Phase 1: The study used the content validity index (CVI) test tool to confirm the factors affecting the acceptance of online loan platforms in Thailand. According to the literature review, Gilbert and Prion [62] said that using 5–10 experts is a very reasonable size. Having more than ten experts is considered unnecessary.

Phase 2: We confirmed the factor components obtained in Phase 1 using a questionnaire-based survey of the second sample according to the theory of [63], which states that the

number of participants included in the research should be at least 250. In addition, our research used 13 observed variables calculated according to the principle of [63], which states that the sample size should be 10–20 times the number of observed variables. In this research, this is equal to 260 samples. To conduct an effective investigation, the researchers decided on a sample group of 300 participants.

## 4. Results

P2P Lending is a financial market platform that can be divided into the lender and borrower sides, and P2P lending connects lenders and borrowers. This research presents the lender's perspective because lenders are at risk if an existing P2P lending platform is unreliable.

The research objectives in this chapter show the lender's point of view. The extraction of success factors from the lender's perspective are focused on the risk, trust, and information philosophy, all of which factor into the lender's decision. The factor extraction point of view were taken from the previous literature with a focus on the first quartile (Q1) to gain credibility and confidence when applying the factors to create a practical conceptual model. The developer's role was to create a web application to support research and evaluation and to find a conceptual model. This study is quantitative. The researcher can evaluate the conceptual model using the P2P lending questionnaire that asks financial business experts to provide advice by completing online surveys that explore the perspectives of potential lenders.

The conceptual model includes the mapping of factors affecting P2P lending development, leading to the next phase of the framework. The study looks at lender behavior, the factors of which are reliable and influence the risk of P2P lending management. This research can help us to develop principles and rationales to develop and run a business that stands the test time. Moreover, there must be good data governance.

### 4.1. Conceptual Model Design

We created a conceptual model of P2P lending that incorporates factors from the literature review to explore and map the risk, trust, and information aspects in order to identify success factors in terms of intent to use the peer-to-peer lending platform. The actual outputs are the risk factors (higher interest rates, inflation increases, macroeconomics, regulation laws and legal, liquidity, and financial and credit status), trust (demographic characteristics, biological factors, and individual reputational capital), and information on the lender's perspective (loan delinquencies, funded loans, politics, and culture).

The key factors were gleaned from the literature review. We prioritized the critical factors and took the variables from financial factor reviews. The first factor is the risk; we looked at the relationships between higher interest rates, inflation increases, macroeconomics, regulation laws and legal, liquidity, and finance and credit status and the decision behavior of P2P lenders [14,64]. The second factor is trust; we looked at the relationships between demographic characteristics, biological factors, and individual reputation and the decision behavior of P2P lenders. The last factor is information; we looked at the relationships between loan delinquencies, finances and income, funded loans, politics, and culture and decision behavior of P2P lenders. Furthermore, we aimed to find variables for each factor. These factors are significant and should be considered when looking at peer-to-peer lending platforms [36,42,65].

Figure 3 shows the P2P lending conceptual model design based on the literature review that maps 3 contexts with 13 factors.

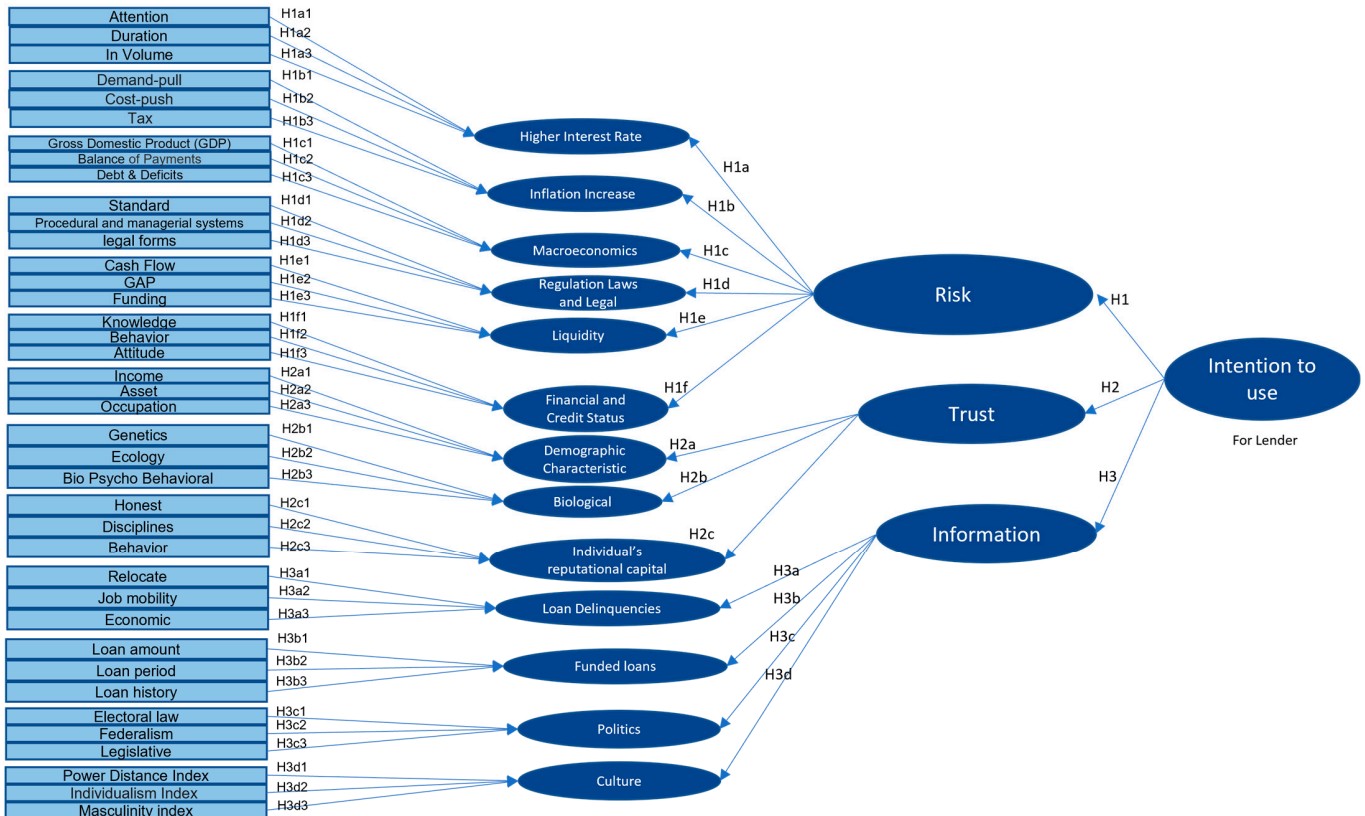

**Figure 3.** Peer-to-peer lending conceptual model design.

### 4.2. Measurement Scale Development

The confirmation factors were based on the content validity index (CVI) of a questionnaire survey with nine experts, one of whom performed a business analysis of P2P lending. The questionnaire used Likert scales. If the number of experts is less than 5, then the CVI value should not be less than 1, and if the number of experts is less than 5–10, the CVI value should be less than the specified value. The researcher will have to review the content to re-evaluate the CVI until an acceptable CVI value is obtained according to the number of experts.

The research uses tools to measure the content accuracy and consistency of the conceptual model objectives extracted by the researchers from the literature review. Initially, the researchers must establish the conceptual model and the questionnaire's accuracy, consistency, and clarity. We conducted a pilot assessment of the questionnaire with experts. We used Cronbach' alpha to the quality and reliability of the research tools. We also used the content validity index (CVI) and index objective congruence (IOC). We propose 3 contexts and 13 factors as variables in the questionnaire are listed in Table 2.

**Table 2.** Hypothesis.

| Factors | Variables | Item/Reference |
|---|---|---|
| Risk (RK.) | higher interest rates (HI) | H1a: The risk factor of a higher interest rate negatively correlates with the decision behavior of lenders who lend in P2P lending [14,18]. |
| | inflation increases (II) | H1b: The risk factor of inflation increase negatively correlates with lenders' decision behavior in P2P lending [16,25]. |
| | macroeconomics (ME.) | H1c: The risk factor of macroeconomics negatively correlates with the decision behavior of P2P lenders [16]. |
| | regulation laws and legal (RL.) | H1d: The risk factor of regulation laws and legal positively correlates with the decision behavior of P2P lenders [26,28,66]. |
| | liquidity (LQ.) | H1e: The risk factor of liquidity positively correlates with the decision behavior of P2P lenders [31,32,67]. |
| | financial and credit status (FC.) | H1f: The risk factor of financial and credit status positively correlates with lenders' decision behavior in P2P lending [35,36,68]. |
| Trust (TT.) | demographic characteristics (DC.) | H2a: The trust factor of demographic characteristics positively correlates with lenders' decision behavior in P2P lending [1,10]. |
| | Biological factors (BI.) | H2b: The trust factor of biological positively correlates with the decision behavior of lenders who lend in P2P lending [25]. |
| | individual's reputational capital (IR.) | H2c: The trust factor of an individual's reputation capital positively correlates with lenders' decision behavior in P2P lending [26]. |
| Lender Perspective Information (LP.) | loan delinquencies (LD.) | H3: The loan delinquency factor negatively correlates with lenders' decision behavior in P2P lending [18,38,69]. |
| | funded loans (FL.) | H4: The funded loan factor negatively correlates with lenders' decision behavior in P2P lending [21,69,70]. |
| | politics (PC.) | H5: The financial and income factor positively correlates with lenders' decision behavior in P2P lending [31]. |
| | culture (CL) | H6: The political factor positively correlates with lenders' decision behavior in P2P lending [43,44]. |

*4.3. Conceptual Model Validation*

We begin the conceptual model validity test from the lender's perspective by reviewing the literature and extracting factors and variables to question the consistency of the lender's views on risks, trust, and lender perspective information. We started with item objective congruence (IOC) validation index [64]. Then, by interviewing experts and reviewing them for content validity, Cronbach's alpha or alpha coefficients are selected for credibility analysis before the questionnaire is distributed to 300 lenders [65] using the principle of calculating the sample from [63]. Next, the researcher analyzes the data using structural equation modeling (SEM), starting with the first-order confirmatory factor analysis (1-Order CFA) to study the relationship between the causal variables [66]. This is based on structural equations of diagrams based on hypotheses or theoretical concepts.

Moreover, the research results mainly analyzed and explained the relationship between independent variables [23,37]. Affecting the dependent variable in both the magnitude and direction dimensions can describe the relationship directly and indirectly. The correlation coefficient between the variables is analyzed according to the direction of the second-order-confirmatory factor analysis (2-Order CFA). The result of the 2-Order CFA tests the validity of the structure or hypothesis, indicating whether the collected empirical data support the hypothesis.

### 4.4. Reliability Analysis

We use Cronbach's alpha model reliability testing to determine the reliability based on model reviews and modeling. It is divided into four levels: Excellent is 0.9 or higher, High is 0.7 to 0.9, Medium is 0.5 to 0.7, and Low is 0.5 or lower. The 13 structures can be divided 39 items, as shown in Table 3.

**Table 3.** Test of reliability statistics for the pilot study.

| Dimensions | Measured Factors (10) | Items | Cronbach's |
|---|---|---|---|
| Risk | Higher interest rates | 3 | Alpha 0.755 |
| | Inflation increases | 3 | 0.737 |
| | Macroeconomics | 3 | 0.631 |
| | Regulation laws and legal | 3 | 0.681 |
| | Liquidity | 3 | 0.753 |
| | Financial and credit status | 3 | 0.743 |
| Trust | Demographic characteristics | 3 | 0.736 |
| | Biological factors | 3 | 0.772 |
| | Individual's reputational capital | 3 | 0.720 |
| Information | Loan delinquencies | 3 | 0.826 |
| | Funded loans | 3 | 0.733 |
| | Politics | 3 | 0.644 |
| | Culture | 3 | 0.738 |

The results show the value of Cronbach's alpha. For higher interest rates (HI, the calculated value is 0.755. For inflation increases (II), the value is 0.737. For macroeconomics (ME), the value is 0.631. For regulation laws and legal (RL), the value is 0.681. For liquidity (LQ), the value is 0.753. For financial and credit status (FC), the value is 0.743. For demographic characteristics (DC), the value is 0.736. For biological factors (BI), the value is 0.772. For individual's reputation (IR), the value is 0.720. For loan delinquencies (LD), the value is 0.826. For funded loans (FL), the value is 0.733. For politics (PC), the value is 0.644. For culture (CL), the value is 0.738.

### 4.5. Data Collection and Response Rate

This study selects a sample group that affects the decision to lend money. The nature of investors focuses on those with a regular income [68]. This comprises government officials, state enterprises, and employees, including entrepreneurs. Then, we collect data for quantitative purposes by conducting online surveys via Google Forms for national territories.

### 4.6. First-Order CFA Model

The results of the three 1-Order CFA models involve 3 latent variables, risk (RK), trust (TT), and information (IF), from the 1-Order CFA model with empirical data. A good indicator of model fit is accepted when using 13 indices ($p$-value = 0.341; TLI = 0.978; CFI. = 0.984; RMSEA = 0.041; SRMR = 0.079). See Table 4.

The 1-Order CFA model indicates that all factors are within acceptable ranges. Level, which is the risk, has the most significant impact on the P2P lending platform, followed by information and trust (with correlation coefficients of 1.00, 0.913, and 0.818, respectively). The details of the measurement model of intention to use are shown in Figure 4. Therefore, the first-order factor model is based on a good indicator of model fit.

**Table 4.** The result of the confirmatory factor analysis.

| Latent | Risk | | Trust | | Information | | $r^2$ |
|---|---|---|---|---|---|---|---|
| **Observe** | β | SE. | β | SE. | β | SE. | |
| Higher interest rates | 1.00 | 0.042 | | | | | - |
| Inflation increases | 0.935 | 0.063 | | | | | 0.748 |
| Macroeconomics | 0.926 | 0.029 | | | | | - |
| Regulation laws and legal | 0.896 | 0.067 | | | | | - |
| Liquidity | 0.819 | 0.089 | | | | | 0.64 |
| Financial and credit status | 0.921 | 0.043 | | | | | - |
| Demographic characteristics | | | 1 | - | | | - |
| Biological factors | | | 0.653 | 0.164 | | | 0.696 |
| Individual's reputational capital | | | 0.813 | 0.205 | | | 0.952 |
| Loan delinquencies | | | | | 0.778 | 0.096 | 0.606 |
| Funded loans | | | | | 0.96 | 0.08 | 0.922 |
| Politics | | | | | 0.864 | 0.095 | 0.746 |
| Culture | | | | | 0.919 | 0.08 | 0.845 |

| Latent | Intention to use | | $r^2$ | AVE | CR | |
|---|---|---|---|---|---|---|
| | β | SE. | | | | |
| Risk | 1 | 0.062 | - | 0.394 | 0.795 | |
| Trust | 0.818 | 0.092 | 0.603 | 0.267 | 0.486 | |
| Information | 0.906 | 0.055 | 0.855 | 0.322 | 0.65 | |

(*p*-value = 0.341; TLI = 0.978; CFI. = 0.984; RMSEA = 0.041; SRMR = 0.079).

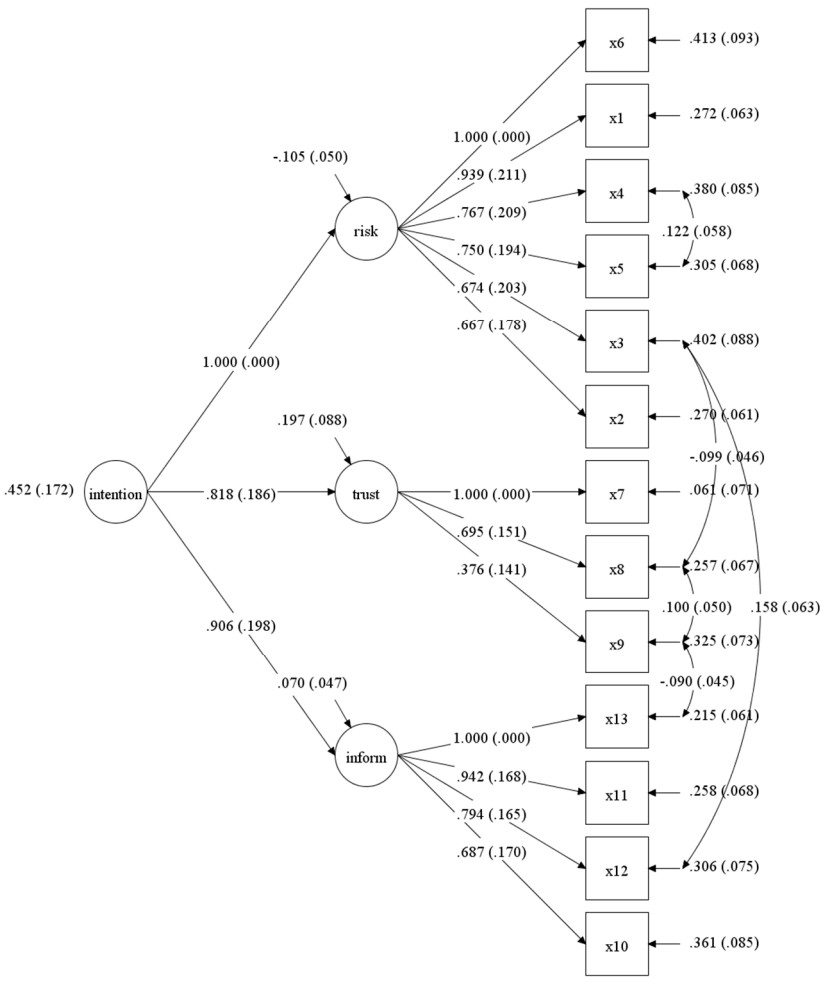

**Figure 4.** The model of CFA.

*4.7. Second-Order CFA Model*

Table 4 shows the results of the 2-Order CFA analysis of intention to use M-Plus version 7.0; the model conformance indexes are as follows: *p*-value = 0.341; TLI = 0.978; CFI = 0.984; RMSEA = 0.041; SRMR = 0.079. The goodness of fit statistics are: sample size > 250, *p*-value > 0.05, TLI > 0.95, CFI > 0.95, RMSEA < 0.07, and SRMR < 0.08.

The conclusion is that the intention to use consists of 3 components: risk, trust, and information. The risk components are measured from six observed variables: higher interest rates, inflation increases, macroeconomics, regulation laws and legal, liquidity, and financial and credit status. In addition, three observed variables measure the component of trust: demographic characteristics, biological factors, and individual reputational capital. The information component is measured with four observed variables: loan delinquency, funded loans, politics, and culture. Considering the weights of the latent variables, the standard weight coefficients were found to be the most critical risk, followed by information and trust; their average weight standard scores were 1.00, 0.913, and 0.836, respectively.

## 5. Discussion of Results and Study Limitations

This study examines the factors influencing the use of a P2P lending platform from the perspective of lenders in Thailand in the form of a conceptual model. First, the research identified the risk, trust, and information factors to make informed decisions about choosing and using an online business P2P lending platform. Lenders need to know and understand the process of preparing a P2P lending platform. We used a literature review to isolate factors that affect lenders' decisions when choosing a P2P lending as they are the first stakeholders. Using this knowledge, we developed a web application to create an efficient P2P lending platform. This information can reduce the risk, increase the credibility of the lender's investment, and result in those involved in the business of P2P lending being able to use the knowledge gained to develop financial work by increasing their financial literacy.

The risk perspective is an essential issue to consider when borrowing money. The key variables include higher interest rates, inflation increases, macroeconomics, regulation laws and legal, liquidity, and financial and credit status. These factors negatively correlate with lenders' decision behavior when it comes to P2P lending. These factors can be considered for lenders in terms of to affect lenders' returns. Moreover, external factors that result in risks to P2P lenders should be understood. Lending online can reduce the risk of unfair interest rates being too high. The law in Thailand and the Bank of Thailand have approved online loan forms, and this macroeconomics mechanism directly affect the ability of lenders to assess lending risk accurately.

The variables affecting the confidence factor include demographic characteristics, biological factors and individual reputation, which positively correlate with the decision behavior of P2P lenders. In addition, perspectives on age, sex, education, and socioeconomic status directly affected lenders trust in the as well as the borrower's genetic traits.

The last data factor used in P2P lending and adoption decisions is, which includes loan delinquencies, funded loans, politics, and culture. In consideration of debt settlement support agencies with which borrowers are involved in transactions, loan delinquencies and funded loans negatively correlate with the decision behavior of P2P lenders, and politics and culture positively correlate with the decision behavior of P2P lender.

In an ever-changing environment, Thailand's financial industry cannot grow without information technology. People's use of digital finance is vital to the country's progress because it not only facilitates easy access to financial information but also affects the gross domestic product. The purpose of this research is to examine the success factors of P2P lending for the continued expansion and growth of the financial industry in Thailand. from relevant documents, the success criteria were assessed through surveys. Thirteen existing variables were used in this regard. A total of 300 P2P lenders a sustainable situation. An audit found that customer-centricity, mobility, and security management were the top priorities, followed by scalability, innovation, low margins, and compliance implicit.

For the Thai economy to grow continuously, the financial industry must understand and cultivate a risk-free transaction environment, which will allow the financial sector to survive by increasing the self-confidence of consumers. Additionally, we should convince people to apply the new technologies involved. This research provided lenders with information on which elements need to be emphasized or hedged to build business operators' confidence in P2P lending and to obtain information from on external factors to support business decision-making.

This study begins by evaluating the success indicators of P2P lending for emerging countries; therefore, it may be subject to some sampling bias. This method can be applied to developing and developed countries' financial and non-financial industries. For education, future work should assess indicators of success. P2P lending has various benchmarks. This research uses a quantitative method to determine success indicators. In addition, combining qualitative and quantitative research methods will improve our ability to clarify the essential success factors for sustainable business development. This field of research can also be extended to international borrowing.

## 6. Conclusions

This study identified factors related to P2P lenders' decisions and inserted them into a conceptual model. The objective was to determine their consistency with the content validity index (CVI) of a questionnaire survey with nine experts that involved 13 factors with a mean value of 1.00. The model's test results regarding the latent variables are divided into three parts. First, the empirical data measure the latent variables of risk factors. We explored factors correlating with P2P lenders' decision behavior. Higher interest rates have the most significant impact on risk of all six factors considered, followed by inflation increases, macroeconomics, financial and credit status, regulation laws and legal, and liquidity, respectively. Next, we explored the trust factors that correlated with the decision behavior of P2P lenders. Demographic characteristics have the most significant impact, followed by the individual's reputational capital and biological factors, respectively. Finally, we looked at the effect of information factors on P2P lenders' decision behavior. Of these, funded loans have the most significant impact, followed by culture, politics, and culture, politics, and loan delinquencies, respectively. For these three-dimensional factors, we used a first-order CFA model and a second-order CFA model to evaluate the conceptual model's structure.

**Author Contributions:** Conceptualization, N.C. and S.K.; methodology N.C. and S.K.; validation, N.C. and S.K.; formal analysis, N.C.; investigation, N.C.; resources, N.C.; data curation, N.C.; writing—original draft, N.C. and S.K.; preparation, N.C. and S.K.; writing—review and editing, N.C. and S.K.; visualization, N.C. and S.K.; supervision, N.C. and S.K. All authors have read and agreed to the published version of the manuscript.

**Funding:** This research received no external funding.

**Institutional Review Board Statement:** The study was conducted according to the guidelines of the Declaration of Helsinki and approved by the Institutional Review Board (or Ethics Committee) of Mahidol University (Protocol code MU-CIRB 2022/299.0311; date of approval 29 December 2022).

**Informed Consent Statement:** Informed consent was obtained from all the subjects and finance experts involved in the study.

**Data Availability Statement:** The data presented in this study are available from the authors upon request.

**Acknowledgments:** The authors would like to thank the advisor team for their support and care. We would like to thank our indispensable mothers, fathers, and brothers, who are role models, for their encouragement towards success. We would also like to thank our colleagues and P2P lending advisors, as well as the team of experts from the Bank of Thailand who provide ideas and methods for organizing the wok.

**Conflicts of Interest:** The authors declare no conflict of interest.

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
