# Peer review of "Success Factors Influencing Peer-to-Peer Lending to Support Financial Innovation"

_sustainability, doi:10.3390/su15054028_

Round 1
Reviewer 1 Report
The level of English is so bad that the paper cannot be read normally and reviewed.
Starting from the abstract itself and reaching the conclusion, the problems in the paper are huge.
There are many incomplete sentences as well as sentences without verbs.
It needs a complete and major reworking and be resubmitted for review.
Author Response
Dear Editor and Reviewer of MDPI Sustainability
Thank you for the opportunity to revise and improve our manuscript. We appreciate your valuable comments and suggestions. We expect our manuscript to have improved upon the suggested revisions.
From the comments and suggestions, we edited our manuscript as follows:
- Edit the manuscript based on the English editing services of the MDPI version (English-edited-60514).
- Complete and major reworking and resubmit for review.
- Complete and rewrite the focus on the research gap aimed at the success of financial innovation.
- Clear state of the factors (risk, trust, and information).
- Clear state of method and fill discussion of results and study limitations.
- Clear the measurement model.
- Declare the factor loading of the model fit.
- Clear the model’s convergent validity.
- Write simple and complete sentences for the background of the P2P lending industry.
- Clear and justified the selection of factors for P2P lending.
- Examine the hypothesis for the methodology of P2P lending.
- Justified Figure and table for better formats.
- Improve significantly in terms of English writing in the paper.
Sincerely,
Supaporn Kiattisin
Information Technology Management, Faculty of Engineering,
Mahidol University, Nakorn Pathom, 73170, Thailand;
Tel.: +66-81-866-4207
Email: supaporn.kit@mahidol.ac.th
Note: We have already edited the manuscript based on the English editing services of the MDPI version (English-edited-60514). Then edited following the comments and suggestions because we misunderstood the manuscript submission procedure, so we did not submit the English edited.

Reviewer 2 Report
There is no clear focus discussion of why the selected indicators are the key factors leading to the sucess of the financial innovation, and what are the current studies on financial innovation, and where are the research gaps you are addressing. The paper needs to be completely rewritten.
The links between the three factors (risk, trust, information), and of these with the leader behaviour intention, are not clearly stated.
The methods are not clearly stated. Missing discussion of results and study´ limitations.
It is not clear if the measurement model is a formative model (as it shown in Figure 4), or it is was design as a reflective model (Figure 5)?
Did authors assess fit of the measurement model from the factor loadings? Figure 5, illustrates that factor loadings were not significant (level 1%, 5%, or 10%).
Did authors find the model´ convergent validity for the main constructs (risk, trust, information)? For example, the average variance explained not reached that recommended value in scientific literature.
The document is not clear. Many parts of the text are not understood. References are missing in many parts of the text. It is neeed to move other references.
Author Response
Dear Editor and Reviewer of MDPI Sustainability
Thank you for the opportunity to revise and improve our manuscript. We appreciate your valuable comments and suggestions. We expect our manuscript to have improved upon the suggested revisions.
From the comments and suggestions, we edited our manuscript as follows:
- Edit the manuscript based on the English editing services of the MDPI version (English-edited-60514).
- Complete and major reworking and resubmit for review.
- Complete and rewrite the focus on the research gap aimed at the success of financial innovation.
- Clear state of the factors (risk, trust, and information).
- Clear state of method and fill discussion of results and study limitations.
- Clear the measurement model.
- Declare the factor loading of the model fit.
- Clear the model’s convergent validity.
- Write simple and complete sentences for the background of the P2P lending industry.
- Clear and justified the selection of factors for P2P lending.
- Examine the hypothesis for the methodology of P2P lending.
- Justified Figure and table for better formats.
- Improve significantly in terms of English writing in the paper.
Reviewer #1:
|
No. |
Comments |
Revision descriptions |
|
1
|
We have already edited the manuscript based on the English editing services of the MDPI version (English-edited-60514). |
The level of English is so bad that the paper cannot be read normally and reviewed. Starting from the abstract itself and reaching the conclusion, the problems in the paper are huge. There are many incomplete sentences as well as sentences without verbs. It needs a complete and major reworking and to be resubmitted for review.
|
|
Reviewer #2: |
||
|
1
|
Complete and rewrite focus on the research gap aimed at the success of financial innovation. |
There is no clear focus discussion of why the selected indicators are the key factors leading to the success of financial innovation, what are the current studies on financial innovation, and where are the research gaps you are addressing. The paper needs to be completely rewritten. |
|
2 |
Consider the clear state of the factors (risk, trust, and information). |
The links between the three factors (risk, trust, and information) and the lender’s behavior intention are not clearly stated. |
|
3 |
Clear state of method and fill discussion of results and study limitation. |
The methods are not clearly stated—missing discussion of results and study limitations. |
|
4 |
Clear the measurement model. |
It is not clear if the measurement model is formative (as shown in Figure 4) or if it was designed as a reflective model (Figure 5). |
|
5 |
Declare the factor loading of the model fit. |
Did the authors assess the fit of the measurement model from the factor loadings? Figure 5 illustrates that factor loadings were not significant (level 1%, 5%, or 10%). |
|
6 |
Clear the model’s convergent validity. |
Did the authors find the model´ convergent validity for the main constructs (risk, trust, information)? For example, the average variance explained has not reached that recommended value in scientific literature. |
|
7 |
Edit the manuscript based on the English editing services of the MDPI version (English-edited-60514). |
The document is not clear. Many parts of the text are not understood. References are missing in many parts of the text. It needs to move other references. |
|
Reviewer #3: |
||
|
1
|
Write simple and complete sentences for the background of the P2P lending industry. |
Background of the Study: The background of the P2P lending industry in Thailand is unclear to readers. Authors often made statements with incomplete sentences. Therefore, readers may have an uneasy reading experience and many doubts about those claims. It would be good if the authors could write simple and complete sentences. In addition, it needs to be clarified what the underlying research problem established in this study is. |
|
2 |
Clear and justified the selection of factors for P2P lending. |
Literature Review: The study gap needed to be presented clearly and was drawn out from the review of past studies. What has not been studied for the P2P lending platforms in Thailand needs to be clarified. The selection of factors is also unclear, as no justification is made for each selected factor. |
|
3 |
Examine the hypothesis for the methodology of P2P lending. |
Methodology: No specific well-established model is reviewed/adopted or adapted by the study. The author builds their model with little justification made. Thus, the selection of variables and examining hypotheses could be more questionable. |
|
4 |
Justified Figure and table for better formats. |
Figures and Tables: The tables could be easier to read. It would be good if they could be represented in better formats. |
|
5 |
Edit the manuscript based on the English editing services of the MDPI version (English-edited-60514). |
Language: It is highly advised that the authors improve significantly in terms of English writing in the paper. |
Please do not hesitate to contact us if you have any comments or suggestions regarding the revision of our manuscript. We believe that our manuscript would be great upon your comments and suggestions. We look forward to hearing from you soon.
Sincerely,
Supaporn Kiattisin
Information Technology Management, Faculty of Engineering,
Mahidol University, Nakorn Pathom, 73170, Thailand;
Tel.: +66-81-866-4207
Email: supaporn.kit@mahidol.ac.th
Note: We have already edited the manuscript based on the English editing services of the MDPI version (English-edited-60514). Then edited following the comments and suggestions because we misunderstood the manuscript submission procedure, so we did not submit the English edited.

Reviewer 3 Report
The following are my comments on possible improvements made to the paper:
Background of the Study: The background of the P2P lending industry in Thailand is unclear to readers. Authors often made statements with incomplete sentences. Therefore, readers may have an uneasy reading experience and many doubts about those claims. It would be good if the authors could write simple and complete sentences. It needs to be clarified what the underlying research problem established in this study is.
Literature Review: The study gap needed to be presented clearly and was drawn out from the review of past studies. What has not been studied for the P2P lending platforms in Thailand needs to be clarified. The selection of factors is also unclear, as no justification is made for each selected factor.
Methodology: No specific well-established model is reviewed/adopted or adapted by the study. The author builds their model with little justification made. Thus, the selection of variables and examining hypotheses could be more questionable.
Figures and Tables: The tables could be easier to read. It would be good if they could be represented in better formats.
Language: It is highly advised that the authors improve significantly in terms of English writing in the paper.
Author Response

(The authors gave the same response as above.)

Round 2
Reviewer 1 Report
The revised version is much better.
Reviewer 2 Report
Thank you for answering the comments and suggestions.
I think that the manuscript has been sufficiently improved to warrant publication in Sustainability.
Reviewer 3 Report
The article has been improved tremendously from the first version. It is much clearer to the readers now.